# Milestones and Current Dilemmas: Evaluation of Sentencing Standardization for Illegal Possession of Drugs in China

**Jia Wu** [1], **Yang Xia** [1,*] **and Apei Song** [2,*]

1    School of Law, Beijing Normal University, Beijing 100875, China; wujia@mail.bnu.edu.cn
2    School of Law, Society, and Criminology, Faculty of Law and Justice, University of New South Wales, Sydney, NSW 2052, Australia
*    Correspondence: yangxia@bnu.edu.cn (Y.X.); apei.song@unsw.edu.au (A.S.)

**Abstract:** It has been more than ten years since the nationwide sentencing standardization reform was implemented in China to solve the widespread problem of uneven sentencing in criminal justice. A statistical analysis of 1595 written judgments of illegal possession of drugs showed that the reform of sentencing for the standardization amount-based crimes has achieved remarkable results, and judges' discretion has been highly normative and consistent. Under the same criminal circumstances, the degree of consistency between the amount involved in the crime and imprisonment has significantly increased, which is more in line with the standards of formal justice. However, the effect of the sentencing standardization reform declined as the amount involved in the crime increased. This exposes the shortcomings of the standardized sentencing model when considering multiple crimes; these include confusion between the amount and circumstances of a crime, the imbalance between crime and punishment, and the application of discretionary circumstances in sentencing depending on the amount involved in the crime. Therefore, it is necessary to attach more importance to the evaluation of the legitimacy of the sentencing range established by criminal law in subsequent sentencing reforms and to further refine and perfect the standardized sentencing mode, with a shift from formal justice to justice in form and substance.

**Keywords:** sentencing standardization; justice; conviction model; illegal possession of drugs; evaluation





## 1. Introduction

Cases can be categorized as simple or complex based on the problematic relationship between applicable legal standards and ongoing proceedings.[1] In the current context of the rule of law, however, with the continuous improvement in the criminal law system, the norms, which are logical premises, have become increasingly sophisticated, and the number of complex cases has also decreased. The low rate of acquittals in criminal prosecution cases[2] also indirectly confirms that the difficulty in most criminal cases lies in conviction and sentencing. As a basic principle of the constitution and criminal law, "equality in the application of the law" requires equal punishment for equal crimes and equal protection for equal victims (Bai 2003).

Since 2010, local Chinese courts have implemented the Supreme People's Court's proposed sentencing standardization reform on an extensive trial basis in response to this demand and to address the common imbalances in sentencing in the criminal justice system. The Opinions on Certain Issues Concerning the Standardization of Sentencing Procedures

---

1    See MacCormick's (2006) book. In simple cases, the judge does not dispute the legal norm that is the logical premise of deductive reasoning, and the validity of the case verdict only needs to be based on proof of logical deduction. Complex cases are those in which the legal norm that is the logical premise is disputed or no corresponding legal norm exists, and the judge needs to create a new rule according to his value judgment.
2    Report on the Work of the Supreme People's Court (SPC): The total number of cases in 2014 was 1,023,000, and the acquittal rate was 4 per 10,000; the total number of cases in 2016 was 1,099,000, and the acquittal rate was 5 per 10,000; the total number of cases in 2018 was 1,198,000, and the acquittal rate was 3 per 10,000.

(for trial implementation) and the Guidelines on Sentencing by the People's Courts (for trial implementation) provide more specific guidance on sentencing methods by improving the correspondence between case elements and sentencing steps, adding more normative elements to the traditional experience-led sentencing model and reducing the arbitrary and capricious elements in judge-imposed sentencing. In 2021, the Supreme People's Court and the Supreme People's Procuratorate issued the "Guidance on Sentencing for Common Crimes (for Trial Implementation)".

The conventional estimative sentencing model in China has been replaced with a standardized one over the last ten years, according to which "qualitative is the major focus, and quantitative is supplemental."[3] However, there are questions concerning the viability of the sentencing guidelines in trials as sentencing norms continue to be improved. What specific function do the various components of standardized sentencing serve at the normative and practical levels? Is it possible for the normative components of sentencing to have the desired effect of inducing restraint in judges' sentencing practices, and is this normative effect of judicial reform long-lasting or only temporary? Are there any more "invisible facts" (J. Zhang 2020) that have a minor influence on judges' sentencing? Does the quality of a sentence's balance improve with the level of detail provided in the norm? All of these are issues that should be left to scientific study.

Standardized sentencing has received attention because guidelines can be used to achieve sentencing consistency (Pina-Sánchez and Linacre 2013), which is one of the fundamental principles in establishing a fair legal system. For instance, the Sentencing Guidelines Council in England has created offense-specific guidelines to inform judges of the need to consider pertinent elements, such as the gravity of the offense, the existence of aggravating and mitigating circumstances, and the possibility of entering a guilty plea (see Sentencing Guidelines Council 2005). The process of normative sentencing has been studied, with a focus on the degree of variation in sentencing outcomes across courts (Britt 2000; Pina-Sánchez and Grech 2017), as well as on the importance of a localized sentencing culture and the roles of resources and caseloads (Dixon 1995; Kautt 2002; Church 1982). In addition, it is important to consider more expansive geographic units, such as counties (Fearn 2005) and districts (Feldmeyer and Ulmer 2011; Johnson et al. 2008), as well as the socioeconomic and demographic traits within districts. In order to construct normative sentencing practices and achieve consistency, it is necessary to summarize intra-court variations (Anderson and Spohn 2010), consider regional, cultural, and national variances, and refer to key characteristics of similar instances (Kautt 2002).

Research on sentencing patterns has shifted from normative to empirical analyses with the progressive formation of China's normative sentencing approach in the form of judicial interpretation in trials. With the use of abstract typologies of actual court difficulties, academics have been able to define the circumstances or issues surrounding the sentencing for offences in an objective manner (Chen 2016; Yu and Guo 2014). According to Chu and He (2019), examining the variables that affect judges' sentencing or examining the impacts of normative elements on the actual control of sentencing are the options here, whereas the goal of empirical methods is to "describe the effects of the administration of criminal policy" (Wu 2021; G. Wang 2020; Zhou 2021; Mo 2015).

However, many empirical studies on sentencing practices in Chinese law still end at the descriptive stage, focusing on abstract typologies of actual legal issues and different kinds of descriptive statistics, which can only result in an objective statement of the facts surrounding the current sentencing environment for a particular crime. It is still necessary to investigate and test the causes and effects. Extracting general sentencing issues from the facts of sentencing is difficult because the selection of individual offenses is too obvious

---

[3]　The estimation sentencing model means that the judge, without strictly distinguishing between the circumstances of the conviction and the circumstances of the sentence, arbitrarily sentences the offender within or below the statutory limits based solely on his or her sentencing values, general knowledge, and trial experience. The standardised sentencing model is based on the national sentencing guidelines issued by the Supreme People's Court: "qualitative in focus and quantitative in support".

and qualitative. Furthermore, the research process neglects the interaction between the findings of empirical research and sentencing models and instead concentrates on explaining how the rules work or investigating one of the two, without eventually returning to the sentencing technique itself. Fragmentation is a defining characteristic of Chinese research as a whole. The sociology of law and the economics of law have not yet been distinguished. It is difficult to have a productive conversation about the philosophy of criminal law. Despite the empirical nature of this study, the objective is to analyse the quantitative causes of uneven sentencing or unfair sentencing and test the effectiveness of the standardized sentencing model for amount-based offenses, but not to provide a comprehensive description of the current state of sentencing for a specific offense.

Therefore, judges' sentencing procedures must be largely reconstructed by using the cases' facts and the verdicts of trials. The crime of illegal drug possession was chosen as a sample crime for observation following a comparison of the 11 different amount-based offenses mentioned in the Sentencing Guidelines on Common Crimes.[4] The choice of the illegal possession of drugs as the offense was made because it is more uniform than the other offenses in terms of the number of sentencing circumstances that can be found, the calculation of the sentence benchmarks, and the offence itself. This also implies that judges will have an easier time excluding the interference of personal subjective evaluations in the crime of illegal drug possession and achieving formal justice by sentencing categorically.

The criminalisation of drug possession began in 1985. China acceded to the international drug conventions and became a party (Sinha 2001; Paoli et al. 2012). The offence of illegal possession of drugs, created in 1990, involved the offences of the possession, purchase, and sale of drugs and was a legislative response to serious drug crimes. It shaped the logic of jail for possession (Carstairs 2006). In 1997, the amendment of the Criminal Law created this offence by obstructing the order of social administration and provided for a three-band sentencing, which continues to this day. Moreover, in 2017, due to the sentencing reform, the sentencing norms for the illegal possession of drugs were refined and the corresponding sentencing starting points were established within the three statutory sentences. To date, the illegal possession of drugs has been known to have the third highest conviction rate for drug offences in China (see Supreme People's Court 2018). Although some Chinese criminologists have engaged in broadly value-based debates on the overall development of illegal drug possession sentencing from institutional, policy, and entitlement perspectives (Lu and Liang 2008; Zhao 2020; Zhang 2011; Wang et al. 2022), few researchers have called for a specific exploration of this offence's developments and shortcomings in the context of a normative sentencing model.

Therefore, we chose the illegal possession of drugs as a lens for balanced sentencing, evaluating the effectiveness of the application of the standardized sentencing model, exploring the dilemmas and reasons for amount-based criminal sentencing, and ultimately suggesting optimal reconfigurations.

## 2. Method

### 2.1. Normative Analysis of Sentencing Models

The comprehensive sentencing guidelines' typical sentencing model[5] generally consists of four steps that closely match each case's elements. Initially, the statutory punishment

---

4  The 11 types of common crimes include theft, fraud, robbery, embezzlement, extortion, smuggling, trafficking, the transportation and manufacturing of drugs, unlawful absorption of public deposits, fundraising fraud, credit card fraud, contract fraud, and the illegal possession of drugs. The 12 types of non-amount crimes include traffic offences, intentional assault, rape, illegal detention, robbery, obstruction of public service, mob fighting, provocation, concealing or concealing the proceeds of a crime, dangerous driving, allowing others to take drugs, and inducing, allowing, or procuring prostitution.

5  In 2021, the Supreme People's Court and the Supreme People's Procuratorate issued the Sentencing Guidance on Common Crimes (for Trial Implementation), which amended the Sentencing Guidance on Common Crimes revised in 2017, while the basic steps of the standardised sentencing model therein remained substantially unchanged. The Sentencing Guidance on Common Crimes (II) of the Supreme People's Court continues to be effective.

establishes the baseline for sentencing based on the essential elements of the crime. Second, the base sentence is established as the starting point for sentencing based on the specifics of the offense, such as the frequency or quantity of offenses, as well as the consequences. The first two steps in an abstract sentencing process for an individual offense, which include creating a benchmark punishment based on the facts of the specific crime, were formulated as clear and comprehensive rules for 23 typical crimes. Furthermore, the base sentence can be modified based on the sentencing factors; typical factors, such as surrender, confession, voluntary confession in court, merit, recidivism, and previous convictions, are modified by setting a percentage increase or decrease, and exceptional statutory sentencing factors are only added to modify the base sentence for nine common crimes. However, the precise percentage increase or decrease is no longer predetermined for each unique sentencing scenario.[6] For cases involving several sentencing conditions, courts have also adopted the strategy of "adding together and subtracting in the opposite way." Fourth, the circumstances of the entire case are considered while determining the penalty.

To match the divisions in Article 348 of the Criminal Code, the three ranges of beginning points are first separated into categories based on the quantity of drugs and the intensity of the circumstances, with the offense of illegal possession of narcotics being used as an example (as in Table 1). The base sentence is then established by raising the severity of the punishment in accordance with additional criminal circumstances that have an impact on the nature of the offense, such as the quantity of drugs involved. Nevertheless, no unique statutory sentencing factors are developed to modify the base sentence once it has been established.

**Table 1.** Comparison between the starting point and the statutory penalty range.

| Classification | Criminal Law Penalty Ranges | Sentencing Guidance Starting Point Range |
|---|---|---|
| The comparatively large quantities of drugs (CLQDs) | Up to three years of imprisonment, detention, or control | Less than one year of imprisonment, detention |
| Aggravating circumstances | Three to seven years of imprisonment | Three to four years of imprisonment |
| Large quantities of drugs (LQDs) | More than seven years of imprisonment | Seven to nine years of imprisonment |

Additionally, seven provincial High People's Courts have improved the sentencing guidelines for the crime of illegal drug possession based on the Sentencing Guidelines on Common Crimes and the Sentencing Guidelines on Common Crimes (II) by considering local customs. These documents are included in the local judicial records (as shown in Table 2). All these provinces have refined and added to the normative components. For instance, all seven provinces improved the quantitative relationship between the amount of drugs used and the corresponding base sentence, thus establishing a quantitative pattern of year-over-year increases in the amount of drugs used and the results of sentencing within the various sentencing ranges created by the statutory sentences (see Table 3). A thorough definition of the phrase "aggravating circumstances" was provided. To provide for aggravating and mitigating circumstances in the crime of illegal drug possession, exceptional sentencing conditions have been included in the majority of provinces. Only Shandong Province has altered the three starting points for crimes involving illegal drug possession and has added a fourth starting point under the heading of "large quantities of

---

[6]   The setting of sentencing circumstances refers to the situation in the Sentencing Guidelines in which exceptional sentencing circumstances unique to a crime are set following the elements of the crime to adjust the base sentence, including intentional assault, rape, illegal detention, theft, robbery, obstruction of public service, smuggling, trafficking, the transport and manufacture of drugs, dangerous driving, and inducing, tolerating, and procuring prostitution.

drugs".[7] The other provinces have only enumerated and refined the various types of drugs that are included in the quantitative classification of "comparatively large quantities of drugs" (CLQDs) and "large quantities of drugs" (LQDs), but they did not include changes to or refinements of the statutory penalty range. The correspondence between the amount of drugs and the sentence according to the benchmark in each province's sentencing laws (as shown in Table 3) is generally under a uniform standard, even though there may be regional variations, especially in the sentencing range of "comparatively large quantities of drugs (CLQDs)", but this demonstrates a high degree of consistency.

**Table 2.** Provincial refinement of sentencing guidelines for illegal drug possession offences.

| Regions/ Provinces | Enumerating the Different Types of Drugs | Refinement of the Relationship between the Amount of Drugs and the Base Sentence | Definition of "Aggravating Circumstances" | Creation of Aggravating Circumstances | Creation of Mitigating Circumstances | Specific Application of Fines | Conditions for the Application of Suspended Sentences |
|---|---|---|---|---|---|---|---|
| Hu Bei | 1 | 1 | 1 | 1 | 1 | 1 | 1 |
| Liao Ning | 1 | 1 | 1 | 1 | 1 | 0 | 0 |
| Si Chuan | 1 | 1 | 1 | 1 | 1 | 1 | 1 |
| Tian Jin | 1 | 1 | 1 | 1 | 1 | 1 | 1 |
| Shang Hai | 1 | 1 | 1 | 1 | 1 | 0 | 0 |
| Shan Dong | 0 | 1 | 1 | 1 | 1 | 0 | 0 |
| Zhe Jiang | 0 | 1 | 1 | 1 | 0 | 1 | 1 |

Note: "1" indicates the existence of this type of refinement; "0" indicates the absence of this type of refinement.

**Table 3.** Correspondence between drug quantity and base sentence for illegal drug possession offences by province.

| Regions/ Provinces | Detention/Imprisonment for a Term Not Exceeding Three Years | More Than Three Years and Less Than Seven Years | More than Seven Years of Imprisonment and Detention |
|---|---|---|---|
| Liao Ning | 1 g/1 Mon | 1 g/2 Mon | 10 g/1 Mon |
| Shan Dong | 1 g/1 Mon | 1 g/1 Mon | 13 g/1 Mon |
| Shang Hai | 1 g/1 Mon | 1 g/2 Mon | 10 g/1 Mon |
| Si Chuan | 1 g/1 Mon | 1 g/2 Mon | 10 g/1 Mon |
| Tian Jin | 1 g/1–2 Mon | 1 g/1–2 Mon | 10 g/1–2 Mon |
| Hu Bei | 1 g/1–2 Mon | 1 g/2–3 Mon | 10 g/1–2 Mon |
| Zhe Jiang | 5 g/4 Mon | 5 g/6 Mon | - |

Note: Based on heroin and methamphetamine.

### 2.2. Hypothesis, Sample, and Verification

This study developed the following four hypotheses from the standpoint of stages and results of sentencing to guarantee the accuracy and orientation of this review and to further elucidate the specific meaning of "verification". These are used to build a bridge between legislative objectives and sentencing practice and to describe the dimensions in which the test indicators and their "ideal indicators" are located. The validity of these hypotheses will be tested through a specific analysis of sentencing data for illegal drug possession offenses in a particular region.

---

7   The Implementing Rules of the Guidance on Sentencing for Common Crimes (II) of the Shandong Provincial High People's Court (for trial implementation) limit the starting point for the illegal possession of a large quantity of drugs from "imprisonment for a fixed term of less than one year or detention" to "imprisonment for a fixed term of three months to one year." In addition, the starting point for the illegal possession of a large amount of drugs is limited from "seven to nine years' imprisonment" to "seven to eight years' imprisonment". A new provision was added: "For illegal possession of heroin or methamphetamine of 500 g or a large quantity of other drugs, the starting point for sentencing shall be determined within ten to eleven years of fixed-term imprisonment".

2.2.1. Hypotheses

**a. The sentencing will be fairer if the norms are more specific.** Establishing more specific sentencing guidelines will facilitate the transition from informal to formal justice in sentencing and increase the likelihood that similar cases will receive similar sentences. This hypothesis investigates whether formal fairness in sentencing and the level of detail in sentencing rules are compatible with the expected positive association predicted by the legislation.

**b. The sentencing reform's normative effect will continue to influence judges' sentences.** The addition of interpretable judicial materials pertaining to sentences will have a consistent, smooth directing influence on the severity of judges' sentencing (see Hofer et al. 2004). This hypothesis explores if the effect of normative sentencing alters the severity of sentences handed down by judges for the same offense and, if there is a change in the temporal dimension of this effect, it is persistent over time or it simply has a temporary impact.

**c. The normative sentencing model may have the desired restraining effect on the sentencing procedures of judges.** This hypothesis investigates whether the aspects of offense-specific sentence adjustments and the sentencing techniques in the Sentencing Guidelines and Sentencing Guidelines (II) can limit judges' sentencing practices. Specifically, a judge will decide the starting point for the base sentence based on CLQDs, aggravating circumstances, and LQDs. Then, the judge will decide on the base sentence based on the facts that affect the composition of the offense, such as the amount of drugs. Eventually, the base sentence will be changed in accordance with the specifics of the case's sentencing. However, no specific sentencing factors for changing the base term for the crime of illegal drug possession have been specified. The examination of circumstantial factors concentrates on typical sentencing factors, such as confession, surrender, merit, recidivism, and prior convictions. The alternative to this idea is that in the sentencing process, judges "collectively betray" the normative sentencing model, as shown by the pervasive "failure" of some normative guiding aspects in sentencing processes, leading to sentencing outcomes that go beyond the sentencing model.

**d. Some "invisible facts" also slightly affect judges' decisions.** In addition to the sentencing elements addressed by the normative sentencing model, there are policies and other non-statutory factors that have a consistent and steady influence on judges' sentences. This hypothesis investigates whether extraneous case facts consistently affect judges' sentences after controlling for the normative sentencing criteria and whether legal principles govern this influence (see Ulmer 2012; Green 1998).

2.2.2. Sample Selection

In this study, all illegal drug possession cases from 2016 to 2020 in Y province were selected as the data source, which was mainly based on the following three aspects. First, in order to compare illegal drug possession during the two stages before and after the implementation of the Sentencing Guidelines (II), the observation sample's time frame was set to 2016–2020 so that the changes in the normative effect of the normative sentencing reform within the temporal dimension could be taken into account. The effects on sentencing reform for drug possession offenses were compared.

Second, the locational aspect was considered. Seven provinces have already published similar sentencing guidelines for the crime of illegal drug possession: Hubei, Liaoning, Sichuan, Tianjin, Shanghai, Shandong, and Zhejiang. In this study, in addition to these seven provinces, Province Y was chosen as a data source to avoid discrepancies resulting from changes in and adjustments to the normative sentencing model in the sentencing rules between provinces and to maintain the relative consistency of the decisions in the sentencing sample. Moreover, Province Y is situated in the western region of China, close to regions where drugs are imported, and drug-related crimes, such as offenses involving illegal drug possession, have always been common there. The number of illegal drug possession cases in Province Y is significantly higher than that in 75% of the other regions in the nation, suggesting that Province Y has had more trial experience than other provinces

according to the number of first-instance judgments on illegal drug possession offenses nationwide uploaded to the China Judicial Documents website (as shown in Table 4).

**Table 4.** National quartiles of illegal drug possession convictions and Y province convictions.

| Percentile (Number of Cases) | 25th | 50th | 75th | Y Province |
|---|---|---|---|---|
| 2016 | 104 | 181 | 297 | 552 |
| 2017 | 88 | 143 | 337 | 551 |
| 2018 | 62 | 94 | 299 | 548 |
| 2019 | 24 | 61 | 179 | 492 |
| 2020 | 14 | 34 | 101 | 266 |

As a result, the database of Beihang University Fabao (BUF), which contained a total of 2114 first-instance criminal verdicts for illegal possession of drugs in the province of Y from 2016 to 2020 was chosen as the sample source for this study. Judgments with common offenses, multiple offenses, and life sentences, as well as cases with incomplete statistical data, were screened and excluded from the study because its main goal was to demonstrate the process of sentencing by judges. As a result, as few non-essential variables as possible were included in the quantitative reasoning process. The selected examples only included individuals who were over 18 and in illegal possession of narcotics. Moreover, the drug categories were restricted to opium, heroin, and methamphetamine[8] to avoid inaccurate drug quantity estimations caused by the presence of multiple drug types; as a result, 1595 sentences were acquired after the screening.

2.2.3. Verification

It is possible to categorize the four hypotheses that were mentioned above into two groups. Hypotheses a and b may both be viewed as studying the relationship between the degree of sentencing normativity and the degree of the meticulousness of statutory norms in the temporal dimension, because the degree of the meticulousness of sentencing norms in China has continuously improved over time. Meanwhile, hypotheses c and d investigate the discrepancy between judicial sentencing practice and statutory sentencing methods by looking at the specific effects of various sentencing factors over the same period. As a result, two procedures for verifying these four hypotheses must be developed for this investigation.

The first procedure was an assessment of the sentencing trends in the sample cases. There is a certain discontinuity between the sentences pronounced for CLQDs (10–50 g) and LQDs ($\geq$50 g), as shown by the scatterplot of drug possession and sentencing results (Figure 1, left). The increasing pattern of sentencing also demonstrates that when the amount of drugs exceeded 50 g, the increase in penalties was distributed more gradually. NMDs were run on the entire dataset, splitting it into two groups—CLQDs and LQDs— with 50 g as the cutoff (Figure 1, right). The differences between the two subgroups were then further tested by using an analysis of similarity (ANOSIM) (R = 0.9582, p = 0.001).

The legal consequences for drug possession in amounts more than 50 g are determined by criminal law. The statutory penalties for illegal drug possession are CLQDs, "serious circumstances", and LQDs, which are equivalent to "fixed-term imprisonment of fewer than three years, detention, or control", "fixed-term imprisonment of more than three years and less than seven years", and "fixed-term imprisonment of more than seven years", respectively.[9] However, the category of "aggravated circumstances" has a much lower likelihood than that of CLQDs or LQDs. There must be a breach in the statutory sentence

---

[8] In the quantification process, the quantities of different types of drugs are converted into the corresponding quantities of heroin by the relevant legal norms. In contrast, opium and methamphetamine are converted under Article 384 of the Criminal Code. All "quantities of drugs" in the following paragraphs are based on the corresponding quantities of heroin after conversion.

[9] Article 348 of the Criminal Law of the People's Republic of China (amended in 2020).

range. As a result, for the sake of testing the hypotheses, each model in the following model design needed to be split into CLQDs and LQDs.

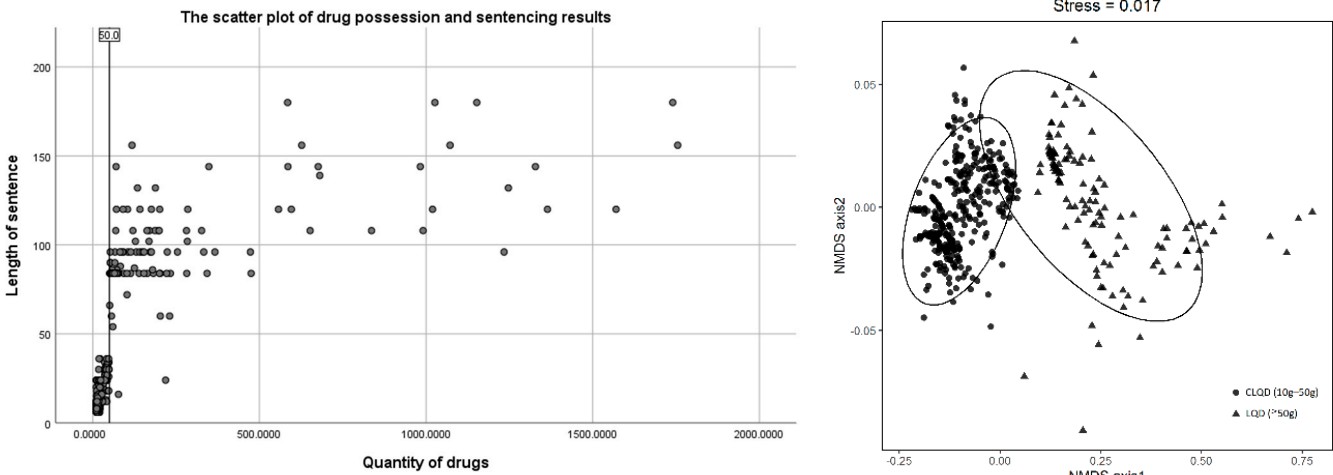

**Figure 1.** Drug quantity and sentence sub-groups.

The second procedure was the design of the test model. Comparing specific sentencing regularization indicators per unit of time was necessary in order to observe the overall trend of sentencing regularization over this period and to demonstrate the temporal effectiveness of sentencing regularization reforms by reflecting changes in the degree of sentencing regularization in the temporal dimension. If the sentencing circumstances of cases are identical, the amount of drugs would ideally be the only variable affecting the sentencing outcomes because the crime of illegal possession of drugs is so widespread that all of the criminal facts affecting the compositions of crimes consist of the amount of drugs possessed. As long as the other sentencing factors are determined uniformly, it is possible to measure the degree of sentencing regularization by comparing the amount of drugs to how close the sentence was to what was intended. By limiting the statutory sentencing circumstances, choosing cases from all of those in the province that met the requirement of having the same sentencing circumstances, and comparing the size of the correlation between the amount of drugs and sentencing outcomes in the clusters of this category of cases in different years (2016 to 2020), a reference indicator of the degree of sentencing normality was used to examine the changes in the normative effect. This rationale was used to guide the study and test hypotheses a and b.

Additionally, a multivariate nonlinear regression model could be created regarding the sentencing provisions of the current legal norms for the crime of illegal drug possession, and the actual trial data from sentencing judgments for the crime of illegal drug possession in Y Province from 2017 to 2020 (the data after the sentencing standardization reform) could be substituted into the model. The influence of each sentencing component of the model and the fit between the data and the model were observed based on the output of R Studio. The force of the "in-law" and "out-of-law" sentencing components was tested, and explanations for the various components influencing judges' sentencing were made. Hypotheses c and d were also tested. The "intra-legal elements" listed in the statute, the "extra-legal elements" existing in the cases, and procedures that could have an impact on sentencing were all taken into consideration when building the multivariate nonlinear regression equation (see Table 5).

Table 5. Variables information.

| Dependent Variable | Explanatory Variables | | | | Control Variables |
| | In-Law Elements | | | Out-of-Law Elements | |
| | Basic Constituent Facts of the Offence | Other Criminal Facts Affecting | Sentencing Circumstances | | |
| Sentence of fixed-term imprisonment | Quantity of drugs | Quantity of drugs | Previous conviction, recidivism or drug recidivism, surrender, confession, merit | Advocacy, suspected drug trafficking, confirmed drug use | "Aggravating circumstances", minor (age), attempt, accessory, type of drug possession |

It is difficult to differentiate among the terms "plea of guilty", "guilty in court", and "confession" if they all appear in the same decision. It was assumed that the model included all three. As a result, only "confession" was used as a condition that was observed in this study, and "plea" and "guilty in court" were not considered for determining punishment. Regarding "previous convictions", "recidivism", and "drug recidivism",[10] since every case involving a prior conviction was identified by the defendant's specific commission for the prior offense, it was possible to distinguish between "previous convictions that do not constitute recidivism or drug recidivism" and "previous convictions and recidivism or drug recidivism", and the fact that these two make up different portions of the sentencing range in the current law makes this distinction clear. Therefore, in this study, we included "previous convictions that do not constitute recidivism or drug recidivism" and "recidivism or drug recidivism" as "previous convictions" in the model.

The adjustment of the base sentence is based on the formula of "add in the same direction and subtract in the opposite direction", in accordance with the classification of the aforementioned variables and the sentencing model for multiple circumstances in one crime; that is, if there are multiple pre-crime and post-crime sentencing circumstances in the same crime, then $A \times (1 \pm a_1 + a_2 + a_3 + \ldots + a_n)$, where $A$ is the base sentence, a is the proportion of the adjustment of the sentencing circumstances, and $\pm$ is determined according to whether the sentencing circumstances are aggravating or mitigating circumstances.

In this way, "internal factors", including the drug amount, prior convictions, recidivism (or drug recidivism), surrender, confession, and merit, can be considered by using the system of regression equations. Depending on how they are incorporated into the sentencing circumstances, the "out-of-law elements" are treated as a discretionary sentencing circumstance B, and positive and negative signs are chosen for them based on their general tendency to be lighter or heavier; then, they are included in the model, and the following is obtained: $A \times (1 \pm a_1 + a_2 + a_3 + \ldots + a_n \pm b_1 \pm b_2 \pm \ldots \pm b_n)$.

Based on the provisions of the criminal law and the Sentencing Guidelines (II) for base sentences, as well as the current local sentencing regulations, a universal algorithm for the base sentence in cases of illegal drug possession offenses can be developed. The base sentence is extended by one month for each extra gram of heroin, methamphetamine, or cocaine in cases in which the amount of drugs is between 10 g and 50 g. So, the base sentence is $A_1 = q_1 + (total\ quantity\ of\ drugs - 10) \times 1$, where the starting point $q_1$ corresponds to a range of "less than one year of fixed-term imprisonment or detention".

When the base sentence is $A_2 = q_2 + (total\ quantity\ of\ drugs - 50) \times 0.1$, where the starting point of the sentencing $q_2$ corresponds to "seven to nine years in prison", it can

---

10     In the crime of illegal possession of drugs, for defendants who constitute both recidivism and drug recidivism due to the same drug offence, the provisions of the Criminal Law on recidivism and drug recidivism shall be invoked in the adjudication documents. However, they shall not be repeatedly punished with heavier penalties when sentencing. The term "recidivist or drug recidivist" refers to recidivism or drug recidivism; no specific distinction will be made.

be concluded that for large quantities of drugs (>50 g of drugs), according to the majority rule in the local sentencing rules, "for every additional 10 g of heroin, methamphetamine, or cocaine, the (base) sentence is increased by one month". The following is how the multivariate nonlinear segmental regression model is built:[11]

$$
Prison\ Term \begin{cases} \begin{aligned} & (q_1 + (Quantity_{drugs} - 10)) \times (1 \pm a_1 \times H_{Surrender} \pm a_2 \times H_{confession} \pm a_3 \times H_{merit} \pm a_4 \times H_{Previous\ conviction} \pm a_5 \times H_{recidivism} \pm b_1 \times \\ & \qquad H_{Defence} \pm b_2 \times H_{Suspected\ trafficking} \pm b_3 H_{Confirmed}\ ) \ (10\ g \le Quantity_{drugs} < 50\ g) \\ & (q_2 + (Quantity_{drugs} - 50) \times 0.1) \times (1 \pm a_1 \times H_{Surrender} \pm a_2 \times H_{confession} \pm a_3 \times H_{merit} \pm a_4 \times H_{Previous\ conviction} \\ & \qquad \pm a_5 \times H_{recidivism} \pm b_1 \times H_{Defence} \pm b_2 \times H_{Suspected\ trafficking} \pm b_3 H_{Confirmed}\ ) \ (Quantity_{drugs} \ge 50\ g) \end{aligned} \end{cases}
$$

Note: Each sentencing circumstance is included in the model as 1 for yes or 0 for no.

## 3. Results: The Effectiveness of the Application of the Standardized Sentencing Model

According to the standards for data control, the full sample must be substituted into the appropriate validation techniques based on the classification of the normative sentencing materials and the establishment of multiple test hypotheses. Then, to make an unbiased assessment of the effects of the normative sentencing reform, the flaws in the model must be identified, and the review's objectives must be accomplished. The researchers present the actual effects of the normative sentencing reform in the vertical dimension of time and the horizontal elements according to the analysis of the data from the quantitative results. **Thus, the final findings and their relations are shown in Figure 2.**

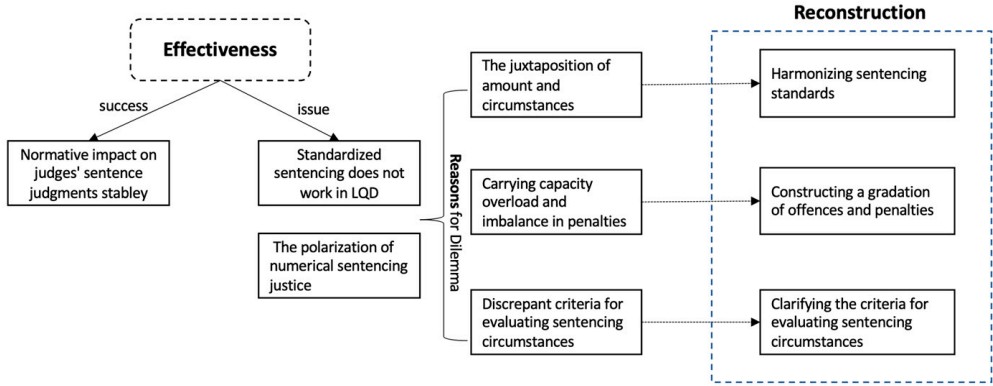

**Figure 2.** The findings' structure.

### 3.1. Sentencing Reform Regulates Stability and Consistency

One cannot simply compare longitudinal trends in the distribution of sentences imposed for illicit drug possession offenses to determine changes in the scope of sentencing for drug offenses because drug crime scenarios are not stable and constant from year to year. However, the crime of the illegal possession of drugs is extremely quantifiable, and if the sentencing circumstances are uniform, it is possible to compare how close the association between the quantity of drugs and the duration of the sentence is. Therefore, the most common sentencing circumstance model ("no previous convictions, lack of surrender, lack of merit, confession on arrival") was chosen as a screening case for uniform sentencing circumstances. The size of the Spearman coefficient was compared between years to examine the changes in the normative effect of the sentencing normalization reform in the temporal dimension.

---

11   As the quantitative relationship between the quantities of drugs and the base sentence is "severed" in the cases of "CLQD" and "LQD", there is a significant difference in the ratio between the quantity of drugs and the base sentence. When including sentencing data, a distinction must be made according to the number of drugs in possession, which is then substituted into the model.

According to Table 6, the Spearman coefficient between the quantity of drugs in one's possession and the term of the sentence from 2017 to 2020 showed a stable increase with respect to 2016 in the sentencing for CLQDs and LQDs, indicating that after the sentencing standardization reform, the link between the quantity of drugs and the length of the sentence was closer under the same statutory circumstances, and formal justice could be better achieved. After the reform, there was a large fluctuation in the quantitative relationship between drug quantity and sentence length. For instance, in the punishment for larger drug quantities, a small amount of difference in the quantitative correspondence was seen in 2019, which was followed by a sharp rebound in 2020. However, overall, since the Sentencing Guidelines (II) were implemented in 2017 and the sentencing standards were further refined, sentences for crimes involving illegal drug possession have become more in line with the formal justice requirements of the rule of law, and the significance of drug quantity in determining the base sentence in such crimes has become clearer. Furthermore, the reform's efficacy has not yet decreased, and its normative impact on judges' sentence judgments has been stable and consistent.

**Table 6.** Correlation testing between drug quantity and sentence term.

| Quantity of Drugs | | Years | | | | |
|---|---|---|---|---|---|---|
| | | **2016** | **2017** | **2018** | **2019** | **2020** |
| Comparatively large quantities of drugs | N | 264 | 283 | 266 | 253 | 90 |
| | Spearman | 0.695 ** | 0.729 ** | 0.792 ** | 0.691 ** | 0.864 ** |
| | sig (One-side) | 0.000 | 0.000 | 0.000 | 0.000 | 0.000 |
| Large quantities of drugs | N | 53 | 81 | 64 | 51 | 28 |
| | Spearman | 0.519 ** | 0.694 ** | 0.665 ** | 0.690 ** | 0.827 ** |
| | sig (One-side) | 0.000 | 0.000 | 0.000 | 0.000 | 0.000 |

Note: $p < 0.01$ (**); Spearman $> 0.5$ (large effect sizes); $0.3 <$ Spearman $< 0.5$ (medium effect sizes); $0.1 <$ Spearman $< 0.3$ (small effect sizes).

*3.2. Some Statutory Sentencing Circumstances Are Ineffective*

The sample cases of illegal drug possession offenses were modelled on CLQDs and LQDs", respectively, after the implementation of the Sentencing Guidelines (II) in 2017. These models were based on the sentencing steps, quantitative relationships, and other case factors recorded in the Sentencing Guidelines and Sentencing Guidelines (II).

For the CLQD phase, the model was immediately applied to all samples from 2017 to 2019. Five samples from the LQD phase were removed because of their extreme drug quantities, which exceeded 3500 g. The results of the two sets of regression analyses were recorded, as shown in Table 7, but the specific coefficients were based on the best fit to the data.

The analysis of the regression model for CLQDs in Table 7 reveals that the R2 of the nonlinear regression model was 0.617, meaning that the model for larger drug quantities was able to explain 61.7% of the variation in sentences for illegal drug possession offenses, suggesting that the statutory model had a more successful fit to the substantive trial data. Only the non-normative elements, such as the defense, had a significant impact on the length of the sentence, while the statutory normative elements, such as the starting point, surrender, confession, merit, prior convictions, recidivism, and drug recidivism, had a significant impact on the sentencing outcome. Regarding these, recidivism and prior convictions were positively correlated with the total sentence length, which is consistent with common sense, while surrender, confession, merit, and defense were adversely correlated with the total sentence length.

**Table 7.** Multiple regression analysis of illegal drug possession offences.

| | Comparatively Large Quantities of Drugs | | | | Large Quantities of Drugs | | | |
|---|---|---|---|---|---|---|---|---|
| | Estimate | Std. Error | *T* Value | Pr(>\|t\|) | Estimate | Std. Error | *T* Value | Pr(>\|t\|) |
| Starting points | 13.24854 | 0.49293 | 26.877 | $<2 \times 10^{-16}$ *** | 101.26794 | 3.39680 | 29.813 | $<2 \times 10^{-16}$ *** |
| Surrender | −0.21590 | 0.03777 | −5.716 | $1.46 \times 10^{-8}$ *** | −0.25713 | 0.04269 | −6.023 | $4.69 \times 10^{-9}$ *** |
| Confession | −0.35624 | 0.01330 | −26.783 | $<2 \times 10^{-16}$ *** | −0.13369 | 0.02396 | −5.581 | $5.10 \times 10^{-8}$ *** |
| Merit | −0.12557 | 0.03626 | −3.463 | 0.000559 *** | −0.26275 | 0.15051 | −1.746 | 0.081810 |
| Previous convictions | 0.07746 | 0.02330 | 3.324 | 0.000921 *** | 0.01608 | 0.03856 | 0.417 | 0.676978 |
| Recidivism (or drug recidivism) | 0.07375 | 0.02616 | 2.820 | 0.004907 ** | 0.12496 | 0.04574 | 2.732 | 0.006647 ** |
| Defence | −0.04391 | 0.01280 | −3.431 | 0.000627 *** | −0.06432 | 0.01841 | −3.494 | 0.000542 *** |
| Suspected drug trafficking | −0.02141 | 0.01264 | −1.694 | 0.090582 | −0.02528 | 0.02329 | −1.085 | 0.278533 |
| Confirmed drug use | −0.02118 | 0.01086 | −1.951 | 0.051353 | −0.02693 | 0.01987 | −1.355 | 0.176354 |

Note: dependent variable: length of sentence for illegal possession of drugs; p < 0.01 (**); p < 0.001 (***).

Specifically, the Sentencing Guidelines (II), which established a sentencing threshold for the range of fixed-term incarceration and detention of less than one year, placed the sentencing threshold at 13.25 months, which was somewhat higher than the value in the model fitting results. Second, all of the sentencing factors related to drug possession offenses during the sentencing process were taken into consideration, and they are listed in descending order of importance as follows: confession, surrender, merit, prior conviction, recidivism, and drug recidivism. This is essentially in accordance with how the Sentencing Guidelines set the mediation ratio for conventional sentencing factors. However, the leniency in situations involving confessions was typically greater—above the typical effect range of 20%—and was situated in the range of reductions for truthful confessions, which was between 10% and 30% of the base sentence for more serious offenses or between 30% and 50% for avoiding terrible consequences.

Finally, among the non-normative factors, only the defense had an impact on the sentence, but less so than the other normative factors. Although other non-normative factors were listed in judgments' language, their effects on the sentences' outcomes were not consistent and uniform. Some texts, such as one concerning "being held in criminal detention by the Public Security Bureau on suspicion of committing drug trafficking (from Criminal Law)", demonstrated that judges strictly adhered to the principle that "suspicion is not a crime" when sentencing for drug offenses, and they did not disproportionately increase the penalties for those who might have been suspected of other drug offenses, since the offense of the illegal possession of drugs serves as the foundational element of drug offenses.

In addition, the defendants also provided the court with some sentencing recommendations, including the following: "The defendant is a drug user, he can honestly confess to the crime after his return to court and has a good attitude towards confessing to the crime so that he can be punished lightly" and "the defendant illegally possessed the drugs for his consumption". To confirm and certify that defendants had illegally acquired drugs, the courts used blood tests, urine tests, access to drug rehabilitation files (such as community drug rehabilitation decisions and mandatory isolation decisions), and relevant administrative penalties (such as administrative penalties for drug use). The courts, however, did not reach a consensus about drug users facing less severe penalties for illegally possessing drugs. The outcomes of this study further demonstrated that "confirmation of drug usage" had no appreciable influence on punishment.

Furthermore, the standard error of the residuals of the nonlinear regression model in the analytical results for the LQD model was 0.201, showing that the statutory model was not built to adequately fit the substantive trial data. Firstly, the suggested beginning point of 101.268 months is roughly 8.44 years, which is in line with the Sentencing Guidelines' sentence range of seven to nine years (II). Secondly, not all sentencing circumstances were

successfully used in sentencing practice. The only statutory factors that had a significant impact on sentencing were the beginning point, surrender, confession, recidivism, and drug recidivism. In contrast, the other statutory sentence circumstances did not have a sufficient regulatory effect on sentencing. The LQD punishment stage was when the standardized sentencing paradigm largely failed. Thirdly, neither "suspected drug trafficking" nor "confirmed drug use" had a substantial effect on the sentencing outcomes, making "defense" the only non-normative factor to do so.

### 3.3. The Polarization of Numerical Sentencing Justice

Figure 3 provides additional comparisons of the nonlinear regression fits of the two models. The actual values are shown in black, and the model predictions are shown in red. The left panel displays the model fit for CLQDs, while the right panel displays the model fit for LQDs.

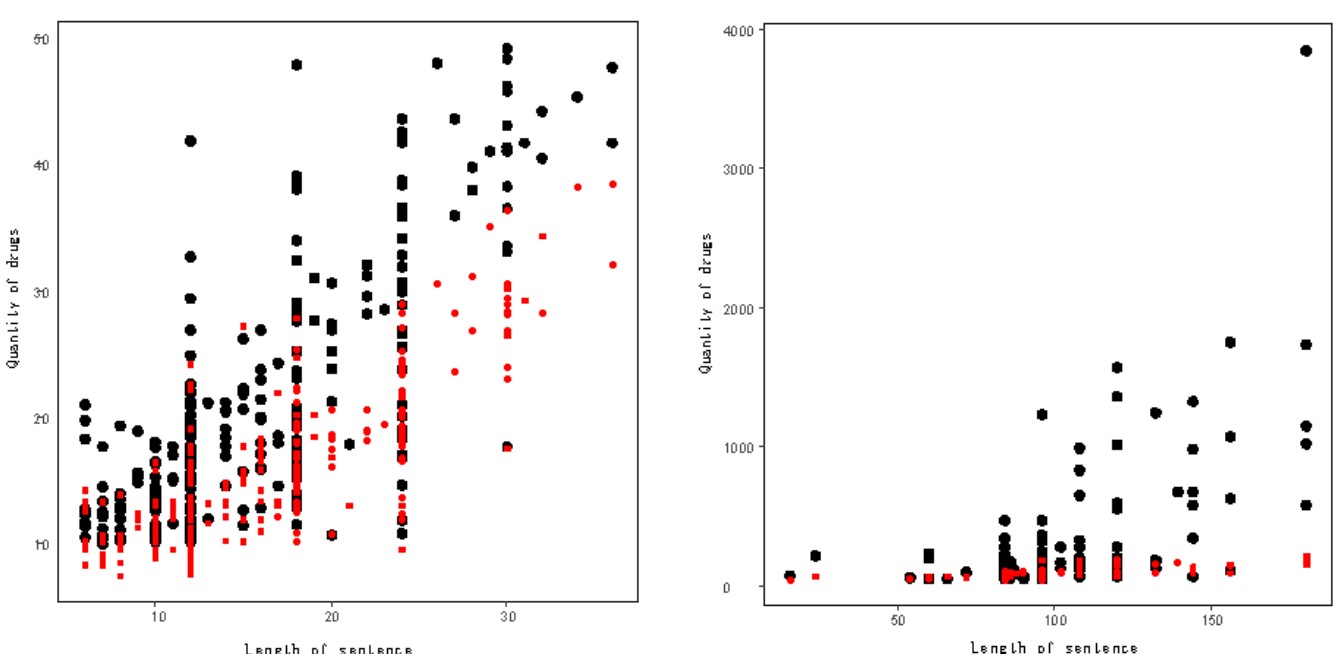

**Figure 3.** Comparison of the groupings of the predicted and actual sentencing values. Red: Sentences predicted according to the non-linear regression model; Black: actual sentences imposed.

The projected and actual values overlapped more in the CLQD model, which had a strong fit, as can be seen in the graph on the left-hand side. The anticipated values in the LQD model on the right (in red) only cover a small fraction of the penalties, with the majority of the offenses exceeding the expected amounts. By comparing the projected values of the model with the actual values of sentences, it was made clear that the penalty gap decreased and tended to converge towards the starting point as the amount of drugs in the defendant's possession steadily grew. The outcomes of cases' punishments became more arbitrary and less distinguished. The mechanism of standard sentencing failed more frequently; the more serious the crime, the more the case's sentencing results veered from the path of formal fairness.

The trial data fit the CLQD model's statutory punishment phases quite well. The cases strictly adhered to the "sentencing by category" requirements, forming a process and a consistent sentencing practice by using the quantity of drugs to establish the starting range of the sentence, using the specific quantity of drugs on this basis to establish the base sentence, and, finally, using the sentencing circumstances to precisely adjust the base sentence. Moreover, the quantity of drugs was utilized as an accurate indicator to distinguish between the instances of categories, allowing for a reasonable increase in the

difference between the sentences imposed and the achievement of a balance between crime and punishment.

The trial data did not match the statutory sentencing steps well in the LQD model, with inflated penalties of seven, eight, and nine years and a flattening tendency in sentencing. The line separating the crimes and the sentences was blurred as the quantities in a defendant's possession increased. Without continuing to pursue exact punishments for the wider range of drug quantities, the sentencing outcomes were confined to the statutory sentence threshold. As a result, there was a departure from the Sentencing Guidelines' tenet that a crime and its punishment must be compatible. As previously indicated, there was a division between the two levels of sentencing precision, even for the "ideal crime" of illegal drug possession.

## 4. Dilemmas in and Reasons for the Application of Standardized Sentencing Models

According to these evaluations of the efficiency of using the normative sentencing model, the level of sentencing normativity was inversely correlated with the level of precise regulation in the vertical temporal dimension. However, the horizontal observations revealed a significant discrepancy between the different sentencing intervals within the same period. The differences in sentences narrowed as the crime increased, and adding more normative components may not improve outcomes. What is the cause of the sentence model's "failure" as more crimes are committed?

### 4.1. Mixed Incrimination Models Compress Sentencing Space

Legal definitions and judicial interpretations of the illegal possession of drugs are consistent with the legislative practice of juxtaposing quantity and circumstances for the majority of amount-based offenses under criminal law. The illegal possession of drugs is a single-amount offense, meaning that the amount is the only factor used to determine whether an offense has occurred. However, the amount is not the only factor used to determine whether the statutory sentence should be increased. As a middle condition for raising the statutory punishment, "aggravating circumstances" were added to the criteria of CLQDs and LQDs.[12]

However, the sample statistics showed that the fraction of cases with "aggravating circumstances" was insufficient, as none of the 1595 cases in the entire sample contained them. The research team also searched the database in China's Judgement Book Network to further clarify the percentage of cases that fit the definition of including "aggravated circumstances". We discovered that since the introduction of the Sentencing Guidelines (II), there have been only 30 cases of illegal drug possession nationwide that fit the definition of including "aggravated circumstances", or 0.15% of all 20,366 first-instance cases.[13] According to the effectiveness of the application of the "aggravating circumstances" standard, which is a separate legal basis for raising punishments, it has not been successful in regulating the results of sentencing. The sentencing regulation paradigm of "amount + circumstances" mostly fails because the "circumstances" part is overly precise and restricts the scope of regulation. In order to guarantee the consistency of "circumstances" in the legal adjudication process, the Supreme People's Court issued the "Interpretation of Several Issues Concerning the Application of Law in the Trials of Drug Crime Cases" in 2016. This interpretation breaks down the "aggravating circumstances" in the second paragraph on the offense of the illegal possession of drugs into three specific acts of possession and a bottom clause on "other aggravating circumstances".[14] Although this process has helped

---

12. The two are in a contradictory relationship. The relationship between two species' concepts with allometric parallelism is contradictory if the sum of their extents is equal to the extent of their genus concepts.

13. https://wenshu.court.gov.cn/, accessed 20 December 2020.

14. Interpretation of the Supreme People's Court on Several Issues Concerning the Application of Law in the Trial of Drug Crime Cases (Fa Shi/Interpretation [2016] No. 8), Article 5: The sentence shall be deemed to be "aggravated circumstances" and the starting point for sentencing shall be determined within the range of three to four years of imprisonment if the illegal possession of drugs reaches the standard of a larger quantity and has one of the following circumstances: if the illegal possession of drugs is in a drug rehabilitation place or a

standardize judicial practice, it has significantly narrowed the types of cases that can be regulated by "circumstances", rendering the function of circumstances as a corrective for the amount unavailable.

Secondly, despite officially raising the punishment range and enriching the classification criteria, the addition of "aggravating circumstances" has narrowed the previous range of sentencing according to the amount. The likelihood of a case having "aggravating circumstances" is much lower than the likelihood of having CLQDs or LQDs based solely on the amount, but this results in a sentence range of up to four years, creating a sentencing gap in judicial practice. In the higher LQD sentencing range, judges typically lowered the sentencing threshold to make sure that the sentences issued in cases that fell just above the range were as close as possible to the statutory threshold for that amount, leaving room for sentencing in situations in which the quantity of drugs exceeded 100 g.

As most offenders suffer from the muddled legislative relationship between the amount and circumstances in terms of sentencing, the problem is not limited to the crime of illegal drug possession. As a result, the judicial issues that arise and the practical solutions provided by judges are strikingly similar. For instance, Amendment (IX) to the Criminal Law establishes an opt-in relationship between the amount and the circumstances in cases of corruption and passive bribery, which also creates a highly complex quandary for the operation of sentencing in situations with competing amounts and circumstances (Y. Wang 2020). In addition, "low-line sentencing" has also contributed to a failure of the ladder effect in sentences for embezzlement offenses. It is necessary to update the range of sentences that apply to each grade. Many judges have a predisposition toward imposing light sentences in cases of fixed-term incarceration due to the lack of a necessary gradient and hierarchies in the harshness of the sanctions. According to Chen (2020), "many cases with divergent amounts of corruption and bribery cannot be brought into line", which is also thought to be the solution to the issue of punishment for corruption and bribery. As a result, uneven sentencing results from the juxtaposition of amount and circumstances not only in the crime of illegal drug possession, but also in other amount-based offenses, making it a general "sickness".

### 4.2. Carrying Capacity Overload and Imbalance in Penalties

To promote formal justice in judicial decisions, the amount in the offense is typically utilized as the primary factor for conviction and sentencing as a means of defining the statutory penalty and determining the beginning point for sentencing. However, in a sample of crimes involving illegal drug possession, it was discovered that as the crime's severity increased, the difference in penalties across instances shrank, and the crime's dominating role decreased, further veering from the normative sentencing track. Some scholars claim that this is because of the legislative model's restrictions on numerical sentencing, which prevent a balanced ladder of correspondence between the "amount" and "amount of penalty", emphasizing that the finite nature of the penalty cannot correspond to the infinite nature of the amount in the offense and that it is impossible to establish a regular and strict proportional relationship between the two (Yu 2016). The inclusion of LQDs in the crime of illegal possession does, in fact, support this issue, but this is not a problem with the legislative model per se, but rather with the level of excellence of the sentencing guidelines and legislative abilities.

Province Y has not established any additional specific sentencing guidelines for the crime of illegal drug possession, in contrast to the other seven provinces that have done so according to the Sentencing Guidelines (II). Instead, in the sentencing range of CLQDs, the statutory penalty's division line indirectly determines the maximum penalty and the maximum amount involved in the crime, thus creating a definite and stable proportional relationship. However, there is no obvious upper limit on the maximum amount in the

---

place of supervision, if the illegal possession of drugs is using or abetting minors, if the illegal possession of drugs is by a state employee, or if other circumstances are severe.

offense in the "large quantity of drugs" sentencing range, and there does not appear to be an increase in the amount in the offense in proportion to the amount of the penalty, which creates a sentencing conundrum with a limited amount of penalty for an unlimited amount of offense. Notwithstanding this, the standardized sentencing system suffers constant harm from the limited number of cases in which the possession of drugs in considerable amounts is, nonetheless, punished with a fixed period of jail.

Only five cases had criminal amounts of more than 2000 g, and 13 cases had criminal amounts greater than 1000 g according to the data from the 1216 samples collected from Province Y after the introduction of the Sentencing Guidelines (II). The criminal amounts in the other cases, however, remained constant at or around 1000 g. The punishment range for situations in which the criminal amount surpassed 1000 g, however, covered ten to fifteen years in jail, which significantly disturbed the relationship between the criminal amount and the base sentence. Judges typically lowered the threshold when sentencing for LQD cases to leave space for cases involving large amounts of drugs. This ensured that the sentences pronounced in cases just above the LQD range were as close as possible to the statutory threshold for that amount, without adding additional penalties based on the circumstances of the case and preventing the sentencing norm system from working.

### 4.3. Discrepant Criteria for Evaluating Discretionary Sentencing Circumstances

In addition to the fundamental imbalance between the crime and penalty amount, some sentencing circumstances need to be better regulated. Based on whether the legislation specifically specifies the content of the sentencing circumstances, they can be split into statutory and discretionary sentencing circumstances (Lu and Zhu 2016). The situations for discretionary sentences are not specifically mentioned in the law and are, consequently, quite open to abuse in judicial practice. "Confirmed drug use" includes the typical characteristics of a discretionary sentencing circumstance in the specific situation of illicit drug possession; it might reflect the social hazard of the defendant's conduct, the threat of danger from the perpetrator, and the level of guilt in the offense (Gao and Ma 2000). In addition, the "confirmed drug use" scenario had a $p$-value of 0.051 in the regression equation, which was practically constant and significant for the sentencing outcomes according to the CLQD model for illicit drug possession offenses.

Nevertheless, in actual sentencing practice, not only are the specific requirements for the use of "confirmed drug use" variable, but the appraisal of the effect is also mixed, as the following discussion of sample instances will show. Drug usage was cited among other statutory mitigating factors in the reasons for decisions in many cases in which "drug use" was considered while sentencing. For example, "the defendant was a drug user and was given discretionary consideration in sentencing (case decision)". Contrarily, there were instances in which "drug use" was negatively assessed as an aggravating circumstance for the crime of illegal drug possession, such as from the perspective of special prevention, when the offender had a difficult time quitting their drug use, and when their likelihood of recidivism was higher than that of other offenders, such as "XX is a drug-addicted person with serious personal danger and is punished severely as appropriate. The sentencing recommendation of the public prosecution is appropriate, and the Court adopts it". The use of enforced drug seclusion was also considered as a form of punishment. Contrarily, drug addiction was noted as a typical example of repetition with the following statement: "The defendant XX has been compulsorily detoxified by the public security authorities' multiple times due to his drug use and is punished harshly as appropriate". Moreover, there was a third option of non-evaluation, which typically took the form of the judge rejecting the defense's mitigation argument (no evaluation) that the defendant was a drug user, such as the following statement: "The defense's mitigating opinion that the defendant XX is a drug user is not accepted by this court".

There were three different tendencies in evaluating the exact circumstances of the same offense. The other two, which had the opposite impact and directly went against the idea of equal punishment for the same offense, substantially weakened the objective of balanced

sentencing if the absence of evaluation was still considered a "discretionary" feature of discretionary sentencing conditions. Additionally, this highlights the limitations of the standardized sentencing paradigm regarding the modification of discretionary sentencing circumstances.

## 5. Discussion: Reconfiguration of the Numerical Sentencing Model

The legal sentencing procedure has already established that sentencing should be an empirical activity in which discretion is employed to provide a thorough value assessment (Shi 2020). The quantitative study of this collection of court cases has increasingly shown the flaws in terms of statutory punishments for crimes, demonstrating that the "logical rationality" provided by "doctrinal reflection" is obviously insufficient. Because of the issues seen in judicial practice, China's criminal policy and even criminal legislation must be adjusted.

### 5.1. Harmonization of Sentencing Standards

The normative purpose of criminal law comprises three requirements. First, criminal law norms should have a fair orientation; second, criminal law norms should be explicit; and third, criminal law norms should be self-consistent, including their harmonization between and within legal provisions. One important criterion for determining whether criminal law legislation techniques are appropriately applied is the assessment of whether they contribute to this normative purpose (Wu 2020). A law must be internally consistent and unified to be considered internally harmonized (M. Zhang 2020). The "circumstances", as a distinct class of statutory enhancement, and the amount-based sentencing scale, however, combine in some amount-based offenses to create a hybrid sentencing model that does not function as intended but instead compresses the continuous sentencing space created by a single amount standard, leading to sentencing imbalances.

It is evident that the "circumstances and amount" incrimination model has successfully addressed the drawbacks of the "amount-only" method of sentencing and has modified the amount standards to a limited extent (Yu 2016). However, it is unclear how the criteria for "circumstances" and "quantity" should be related and whether they should be independent of one another or intersect. This is the solution to the hybrid model's issue, as it limits the range of possible sentences. The amount is one of the broad circumstantial elements in the context of the concept of crime, but at the level of conviction, "not all the punishable elements that make an act a crime are constituent elements; only the punishable elements inherent in a given crime, of a type, are constituent elements" (Machino 1989). Thus, the single sum and the narrowly defined "aggravating conditions" are debatable as evidence. The objective evaluation of whether an act constitutes an offense must take both into account. To turn a qualitative crime into a quantitative sentence, it is necessary to consider the quantitative impact of all the conditions when it comes to sentencing. It is simpler to develop more obvious "crime tips", since the amount in a crime is more measurable and continuous than other sentencing criteria (Shi and Su 2021). It is feasible to distinguish between fundamental and auxiliary criteria by utilizing the amount as a criterion for selecting the standard sentence and the circumstances to modify the base sentence.

When establishing upgraded statutory sentences, one should consider how well the standard of upgraded sentences quantifies crimes to establish a sentencing system in which crimes and penalties are compatible, as well as to achieve the same sentences for the same cases, rather than blindly pursuing a variety of upgraded sentences. This implies that different standards must be developed to quantify offenses. To be equivalently converted at a certain statutory level, each sentencing standard must, however, correspond to a whole system.

Consider the crime of illegal drug possession as an illustration. First of all, the primary factual component of the offense should be the quantity of narcotics in possession. It is necessary to specify the two categories of CLQDs and LQDs, as well as the sentencing ranges for each. In addition, to create a continuous and comprehensive criminalization

standard for the amount, the unit of the crime and punishment corresponding to the second paragraph of Article 348 of the Criminal Law, "severe circumstances", should be eliminated. Secondly, the three specific "aggravated circumstances" are listed in the corresponding sentencing guidelines as special sentencing circumstances for the crime of illegal drug possession. These circumstances are used to modify the base sentence, and a ratio of the specific circumstances has been established. The logical ambiguity between the amount and circumstances in the division of statutory penalties can be cleared up by changing the "aggravating circumstances" from sentencing circumstances to statutory penalties. In addition to achieving the convergence of the "amount" standard in the sentencing scale, the establishment of a unified standard for the division of statutory penalties can also increase the individual sentencing range, thus aiding in the resolution of the issue of uneven sentencing brought on by "pressure line sentencing" and "bunching sentencing" in judicial practice.

### 5.2. Constructing a Gradient of Correspondence between Offenses and Penalties

Based on the harmonization of the various sentencing guidelines, additional attention must be paid to the issue of normative failure brought on by the application of limited penalties to infinite offenses. Clarifying the range of adjustment between the amount and the penalty in each offense and creating a connection between the crime and the sentence duration are the solutions to this issue. In addition, the present judicial rationality in creating accurate sentences proportionate to the crime and punishment has a distinct break. According to the issue of justice, judicial sentencing logic can be categorized into the national, regional, and case-specific categories[15] (Wang 2021). National judicial rationale divides the sentencing range in terms of the normative hierarchy. Regional judicial reasoning, on the other hand, is more tightly tied to situations. To support the regional punishment category, it assumes the role of calculating the precise value of the criminal amount within the regional scope. Regional sentencing guidelines, however, are still in need of urgent revision and updating, and sentencing guidelines for specific cases still mostly rely on the national judicial rationale for the specified range and the judges' own trial experience.

In terms of the number of regulatory bylaws, the Supreme People's Court's Sentencing Guidelines have been implemented by 30 provincial-level administrative regions around the country. However, only three provinces have issued "complete" corresponding implementation rules for the Sentencing Guidelines, the Sentencing Guidelines (II), and the Opinions on Certain Issues Concerning the Regulation of Sentencing Procedures (for Trial Implementation), respectively. Only six provinces have issued corresponding rules for the Sentencing Guidelines (II). Moreover, some provinces have lagged in the implementation of sentencing rules, and even after the national sentencing standards were revised, some invalid sections in local sentencing laws continued to be in effect. As a result, there is a rift and conflict between the regional and national judicial logic, which is especially apparent in the case of repeat offenders, for whom standardization and refinement are urgently required.

How might regional sentencing practices for specific crimes be abstracted by local judicial rationality? Based on the provincial drug crime situation, the ideal outcome is to establish a threshold for the degree of the crime to be carried by the stage of sentencing and to further develop the relationship between the quantity of drugs and the growth of a punishment range for LQDs. It is necessary to rely on the case disclosure system and intelligent justice technology to determine the distribution of the amount of crime

---

[15] The Sentencing Guidance concentrates on the national collective judicial rationality, a summary and sublimation of the rational national experience. At the same time, the various implementation rules formulated by the provincial high people's courts according to the Sentencing Guidance and other rules are a distillation and generalisation of the sentencing practice in each province. Individual judicial rationality is the judicial discretion of judges in individual cases, while individual judicial rationality should respect collective judicial rationality.

committed in the sentencing range, to quantify and analyse the quantitative relationship between criminal law and punishment, and to refine the correspondence in the range to give an accurate picture of drug sentencing in the region. For instance, the drug situation is acute in Province Y, where the courts received 19,177 new first-instance cases of drug offenses between 2016 and 2018, making up more than one-fifth of all criminal cases in the province. It is critical that the general sentencing leniency continues to ensure some level of distinction in the sentencing for various offenses. In 99.18% of the cases involving the 1595 sentences that are now in effect, the amount of narcotics in the defendant's possession was less than 1500 g. Despite the fact that this standard is higher than the typical standard of 1010 g in other provinces, it is possible to choose a maximum amount of drugs in the defendant's possession to be between the two in order to create a perfect quantitative scale of penalties and seek equitable sentencing among provinces.

Meanwhile, other provinces should categorize and analyse all court sentences in each region based on statistical research when formulating sentencing guidelines, clarify the range of variation in the maximum amount of drugs in the defendant's possession during the same sentencing period for the same offense, determine the general correlation between the amount of drugs and the penalty, restore the proportion of adjustment of the base sentence according to the sentencing circumstances, and provide a reference for legislative improvement with the existing sentencing tendencies.

### 5.3. Clarifying the Effect of the Evaluation of Discretionary Sentencing Circumstances

The evaluation standards for sentencing circumstances should be further enhanced after rationalizing the relationship between the amount and the offense. The uniform evaluation of the same discretionary circumstances in terms of their tendency to be lighter and heavier should be included when developing evaluation criteria for discretionary sentencing conditions outside of the specific legislative norms.

Based on the statutory nature of their substance, discretionary circumstances should be evaluated favourably or unfavourably. In the absence of specific legal provisions, discretionary sentencing circumstances are formally stated; however, this does not imply that they lack a legal or policy foundation, nor that the foundation's legality can be confused with the statutory nature of its contents (Lu and Zhu 2016). For instance, the Criminal Law, the Sentencing Guidelines, the Sentencing Guidelines (II), the Interpretation on the Application of Certain Issues in the Trials of Drug Offenses, and other series of laws and regulations do not directly evaluate possession for one's own consumption in the case of "drug addiction" in the offense of the illegal possession of drugs. However, a certain level of drug use can be considered a mitigating circumstance or, at the very least, should not be considered an aggravating circumstance in accordance with the principle of statutory penalties and the spirit of the Minutes of the Dalian Conference and the Minutes of the Wuhan Conference.

However, does this indicate that an unlawful possessor should have the option of having the punishment reduced if they use the drugs for themselves if drug usage is not deemed to be an aggravating element in the case of illegal possession? The answer is no. The Wuhan Conference stated that although the amount of drugs consumed should be subtracted from the quantity of drugs calculated for the offense of drug trafficking, this provision cannot be directly applied to the offense of the unlawful possession of drugs due to the principle of "the heavier, the lighter" (meaning that a lighter offense is determined based on the more serious one). First, the two offenses defend different legal interests. In the case of drug trafficking, the quantity deduction subtracts the portion of the drug that the trafficker solely intends to personally consume and not to sell. This is so because the drug trafficker does not genuinely intend to sell the drug. When it comes to drug trafficking offenses, the crime of illegal possession of narcotics serves as a catch-all, and this includes criminals who cannot be proven guilty of illegal possession due to a lack of evidence. The holder's desire to use narcotics does not lessen the harm that the act of illegal possession causes to the social system. Consequently, self-consumption cannot be used as a factor in

sentencing for the crime of illegal drug possession. Second, the two offenses have distinct beginning points. There is a set threshold for the crime of unlawful drug possession. Drug possession is only criminal if it exceeds the CLQD, which allows for the possession of modest amounts of drugs for personal use. Although the crime of drug trafficking does not have a set minimum amount and is penalized for the entire act, the amount of drugs that the trafficker personally ingests must be subtracted. Because of this, the amount expended by the traffickers themselves is deducted.

While possessing drugs for personal use is not always justified in the crime of illicit drug possession, this does not mean that the rationale cannot be used as a mitigating factor in sentencing. However, to alleviate it, stronger constraints must be added, and the case's unique circumstances need to be considered in the judgment as a whole. For example, Hubei Province defines a mitigating circumstance as "people who suffer from a serious sickness and take medicines to reduce the pain of the condition and are in possession of narcotics".[16] Only the treatment of serious ailments is allowed under new restrictions on the use of medications. While rigorously limiting the requirements for using the discretionary sentencing circumstances so as not to be detrimental to the balance of the sentence, the discretionary sentencing circumstances complement the statutory sentencing conditions to maintain substantive justice.

## 6. Conclusions

This study modelled standardized sentencing for a few particular amount-based crimes of illegal drug possession. It objectively recreated the thought processes of judges in sentencing and identified the "pain points" for single-amount offenders in the standardized sentencing reform by using an empirical analysis to statistically assess the differences between sentencing norms and trial practices.

The data analysis showed a positive correlation between the formal justice of a sentence and how strict China's existing sentencing laws are. The more specific the sentencing guidelines become as they are continuously amended, the more they aid in achieving formal justice. Second, the normative effect supported by the sentencing reform will continuously have an impact on judges' sentencing, resulting in a fundamental change in judges' sentencing patterns and a noticeably higher correlation between the offense's amount and the length of the sentence in cases of amount-based offenses. Thirdly, not all components of normative sentencing can serve the anticipated restraining function. In addition, as crime increases, the high carrying capacity of crime and punishment might result in the failure of some normative components and the intensification of sentencing. Finally, a few discretionary sentencing factors may have minor influences on judges' choices. However, because there is not a single driving principle, their application and impact are unclear.

Nonetheless, the above-mentioned sentencing issues constitute a common "pain point" in the sentencing for single-amount offenders and are not specific to the offense of illegal drug possession. To improve sentencing based on a unified concept of punishment, it is necessary to fundamentally revise and reconstruct the current standardized sentencing model in criminal law. This will allow us to rationalize the inconsistent relationship between the amount and circumstances in sentencing. This is how formal justice in sentencing can be transformed into sentencing justice that equally emphasizes form and substance.

Indeed, the rethinking of imprisonment for possession is not limited to China, but can also be found in Canada (see Carstairs 2006); the international community is constantly working through evidence to provide rational explanations for a certain amount as a start-

---

16   The Hubei Provincial High People's Court's Sentencing Guidelines on Expanding the Offences and Types of Sentences for Sentencing Standardisation (for trial implementation) (2016): "If one of the following circumstances exists, the penalty may be reduced by less than 30% of the base sentence: (1) if the drug content is low; (2) if the person suffers from a serious illness and illegally possesses drugs in order to alleviate the pain of the illness by taking drugs; (3) Pregnant women, breastfeeding women, persons suffering from serious diseases and other special groups of people who are used or forced to participate in drug offences; (4) other circumstances in which the penalty may be reduced".

ing point or ceiling in amount sentencing to avoid gender/racial/economic discrimination and special circumstances (Unnever 1982; Lipp 2003). This cautionary approach favours avoiding another escalation in the war on drugs. Amount-based sentencing must be supplemented with a rich basis in reality between simple numbers; otherwise, arbitrary cut-offs will reduce the significance of punishment for holders of large amounts but overly oppress those with small amounts. Our research thus clearly points to the need for further reflection in the Chinese context, matching amounts and circumstances and creating a gradient of criteria that does not overly compress the space for comparatively large quantities of drugs.

Due to the quantitative analysis used in this study, there are two limitations: The focus on Province Y was, in part, constrained by the regional data, and alternative data possibilities were necessarily lost in the handling of the variables. This makes our deliberations somewhat limited by the representativeness of the data itself.

**Author Contributions:** Conceptualization, J.W.; methodology, J.W. and A.S.; software, J.W.; validation, J.W., A.S. and Y.X.; formal analysis, J.W.; resources, J.W.; data curation, A.S.; writing—original draft preparation, J.W. and A.S.; writing—review and editing, A.S.; visualization, Y.X.; supervision, Y.X. All authors have read and agreed to the published version of the manuscript.

**Funding:** This research received no external funding.

**Institutional Review Board Statement:** Not applicable.

**Informed Consent Statement:** Not applicable.

**Data Availability Statement:** Not applicable.

**Conflicts of Interest:** The authors declare no conflict of interest.

## Abbreviations

| | |
|---|---|
| Comparatively Large Quantities of Drugs | CLQDs |
| Large Quantities of Drugs | LQDs |

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
