# Peer review of "Milestones and Current Dilemmas: Evaluation of Sentencing Standardization for Illegal Possession of Drugs in China"

_laws, 2023_

Round 1

Reviewer 1 Report

Reviewer comments 

Introduction

·       The purpose of the study was to examine the effects of the sentencing standardization reform applied in China 10 years ago, through a statistical analysis of judgments (verdicts) on illegal possession of drugs. While there is a good overview on this topic, there is a lack of information regarding the punishment policy and the legal system in China. Basic information on these issues should be added, to increase the reader's knowledge and understanding.

·       In addition, it is advisable to add a brief comparison between the legal system and legislation in China in relation to drug offenses (on which the research focuses), compared to Western countries, to obtain a broader perspective on this matter.

Method and Findings

these chapters are listed in a detailed and clear manner. However, there is a load of information that makes it difficult to follow the many findings. Therefore, it is advisable to gather the main findings into a consolidated table.

Discussion - Here too, the research findings should be presented while comparing them with the current situation in Western countries, for the purposes to gain a broader perspective of the study findings.

·        Also, a reference to the limitations of the research should be added, as well as applied implications of the research findings.

Author Response

Thanks for your comments, our response is attached. 

Reviewer 2 Report

This is a very well written paper with detailed analyses, well written conclusions and clearly presented findings. Just a few things I would like to see, if this was to be revised:

-Your literature review/contextual discussion is just not explored enough - I feel that a lot more can be and should be said on that front for the sake of the readers ability to understand the context and to situate the article in the existing literature

-Your article feels under-referenced. This will partially be solved by the expansion of literature review, but I feel it could be done in other parts of the article too, namely the methodology.

Author Response

Thanks for your useful comments, we have revised and attached response here.
